# Antiradical Activity of Hydrolysates and Extracts from Mollusk *A. broughtonii* and Practical Application to the Stabilization of Lipids

**DOI:** 10.3390/foods9030304

**Published:** 2020-03-07

**Authors:** O.V. Tabakaeva, W. Piekoszewski, T.K. Kalenik, S.N. Maximova, A.V. Tabakaev, D. V. Poleshyk, L. Proniewicz

**Affiliations:** 1Department of Food Science and Technology, School of Biomedicine, Far Eastern Federal University, 690000 Vladivostok, Russia; 2Department of Analytical Chemistry, Faculty of Chemistry, Jagiellonian University, 30-386 Krakow, Poland; 3Department of Food Technology, Far Eastern State Technical Fisheries University, 69000 Vladivostok, Russiatym1988@mail.ru (D.V.P.); 4Department of Chemical Physics, Faculty of Chemistry, Jagiellonian University, 30-386 Krakow, Poland; leonard.proniewicz@uj.edu.pl

**Keywords:** antiradical activity, hydrolysates, hydrothermal extracts, *A. broughtonii*

## Abstract

The antiradical properties of hydrolysates and hydrothermal extracts of bivalve mollusks (*Anadara broughtonii*) from the Far Eastern Region of Russia and their influence on lipid oxidation in mayonnaise were investigated. The radical binding activity of hydrolysates and extracts of *A. broughtonii* varies from 55% to 89%. The maximum radical-binding activity was observed for acid hydrolysates. The antiradical efficiency of acid hydrolysates is 35%–41% of the BHT (butylhydroxytoluene) index. The antiradical activity depends on the (method of) technological and biotechnological processing of raw materials. Acid and enzymatic hydrolysates and hydrothermal extracts of *A. broughtonii* in mayonnaise slow down the process of oxidation of lipids and hydrolysis of triglycerides. Acid hydrolysates reduce the speed of oxidation and hydrolysis of lipids in mayonnaise more efficiently than the enzymatic hydrolysates.

## 1. Introduction

The study of free-radical oxidation processes is important today due to their proven influence on various metabolic processes of the human body [1,2,3]. The accumulation of free radicals causes lipid peroxidation and dysfunction of the cell membrane, which leads to premature aging of the body, frequent sickness and even malignant tumors [4,5,6]. A quantitative study of the antiradical properties of given substances can suggest the ways in which such substances with antiradical activity are applied in practice. However, up till now, the focus of studies on antiradical activity has been mainly oriented towards vegetable raw materials—essential oils and mixtures thereof, and extracts of medicinal and other plants [7,8,9]. At the same time, raw materials of animal—including marine—origin, and products (of animal/marine origin) that have been subjected to technological and biotechnological modification could also have antioxidants. The biological and pharmacological activities of organic natural compounds of marine origin have been demonstrated by numerous studies [10,11,12,13].

Nowadays, many species of marine organisms that are used for food purposes, in traditional and non-traditional medicine, were also shown to have antioxidant activity and as raw materials for obtaining biologically active additives. Amongst marine hydrobionts, bivalve mollusks are promising sources of biologically active substances with antiradical properties. Substances from bivalves exhibit high antioxidant capacity: Extracts from *Crassula aequilatera*, *Mactra murchisoni*, and *Paphies donacina*, which were reported to scavenge 2,2-diphenyl-1-picrylhydrazyl (DPPH) radicals [14]; *Paphia malabarica* extracts, which were evaluated in relation to DPPH and ABTS^+^ radical [15] crude extracts and fractions of alcohol extracts from *Mactra veneriformis* [16]; and oligopeptides from *Meretrix meretrix* [17].

The clams *Anadara broughtonii*, *Spisula sachalinensis*, *Mactra chinensis*, and others can be found in the coastal zone of the Sea of Japan [18]. The tissues of bivalve mollusks are rich in amino acids and specific carbohydrates [19,20,21,22,23,24,25]. *A. broughtonii* belongs to the family *Arcidae* and is a rather common type of Mollusca, in the *Bivalvia* class. It is found in the upper-sublittoral in subtropical areas of the Asian Pacific, with commercial stocks at depths of 2–15 m. The maximum lifespan of *A. broughtonii* is 65 years; the total mass of individuals varies from 80 to 380 g, and the shell length is 65–80 mm. *A. broughtonii* inhabits muddy and sandy bottoms, burrowing to depths of 10–25 cm. It is found in the Yellow Sea and the Sea of Japan [26].

Acid and enzymatic hydrolysates and hydrothermal extracts are products of technological and biotechnological processing of bivalve mollusks. They are sources of biologically active substances with potential health benefits [27,28].

Their antioxidant properties may be the basis for use as ingredients in lipid-containing emulsion products in order to reduce the rate of oxidation of lipids. Nowadays, various synthetic substances that perform the function of antioxidants are used for lipid stabilization of vegetable oils—potassium diethyl dithiophosphate or 2,4,6-tris-(dimethylamino) phenol or antioxidant mixtures containing BHT, naphthols, propyl gallates, and others. The disadvantages of these stabilizers are that they are not devoid of toxic properties [29]. Natural antioxidants are safer at stabilizing the quality of lipid rich foods [30].

The purpose of the present study was to investigate the antiradical properties of acid and enzymatic hydrolysates and hydrothermal extracts of the soft tissues of the bivalve mollusk *A. broughtonii* and to determine their effect on the oxidation of lipids in mayonnaise.

## 2. Materials and Methods

### 2.1. Material from the Bivalve Mollusk A. broughtonii

The clam *A. broughtonii* ranges from 65 to 80 mm in length, and its mass varies from 80 to 200 g. Molluscs were collected in June, September, and November 2016, and in February and April 2017 from Amur Bay (43060 N and 131440 E), Sea of Japan, in the Primorsky region in Russia (all analyses were done using pooled mollusks—seasonal changes were not taken into consideration in this study). The mollusks from all the sampling months were pooled to one sample of 5 kg. Live bivalve mollusks were transported under refrigeration (+6 °C) to the laboratory within 3 h and sampled randomly for this study. Upon arrival, the clams were manually shucked by cutting the adductor muscle with a knife. The clam juice was removed and the edible portion, constituting 12.46%–14.08% of the total weight of *A. broughtonii*, was collected. The edible portion was then dissected into 4 parts: Muscle, mantle, adductor, and viscera. Muscle and mantle were powdered using a blender (Phillips, Guangzhou, China) in the presence of liquid nitrogen. The samples were packed in a polyethylene bag, sealed and stored at −20 °C until use. The storage time was no longer than 1 month.

### 2.2. Acid and Enzymatic Hydrolysis and Hydrothermal Extraction and Analysis

Hydrolysates and extracts were obtained separately from the mantle and muscle of the mollusk. The frozen tissues of bivalve mollusks *A. broughtonii* were thawed at 4 °C before use, mixed with 6% solution of citric acid at a proportion of 1:1 according to the mass of tissues and acid, then homogenized in a tissue triturator for 2 min, and homogenate was obtained. The hydrolysis reaction was performed at 95 °C for 8 h. The resulting liquid phase was an acid hydrolysate (AH). In order to obtain enzymatic hydrolysate (EH), tissues of *A. broughtonii* were mixed with deionized water, the proportion being 1:2 according to the mass of tissues and water, then homogenized in a tissue triturator for 2 min, and homogenate was obtained. A hydrolysis reaction was performed at 45 °C and pH 7.5 in a shaking incubator at 150 rpm in order to promote activity of enzymes. The proteolytic enzyme Protomegaterin G20x (TU 00479942-002-94, Russia), produced by *Bacillus megaterium* was obtained from OOO PO SibBioPharm (Berdsk, Russia). Protomegaterin is a food-grade enzyme which has a declared proteolytic activity of 800 U/g. The enzyme was stored at 4 °C until it was used for the hydrolysis experiments.

The hydrolysis time was 8 h and the enzyme:substrate ratio was 5:1000. At the end of the hydrolysis, the liquid phase was separated by filtration, and boiled for 20 min in order to inactivate the enzyme. In order to obtain hydrothermal extracts (HTE), *A. broughtonii* tissues were mixed with deionized water in a 1:1 ratio according to the mass of tissues and water, and were then homogenized in a tissue triturator for 2 min, and homogenate was obtained. High-temperature processing was performed at 100 °C, duration 3 h. The moisture content was measured according to methodology described by the Association of Official Analytical Chemists (AOAC) (2000). Samples were dried in an oven at +105 °C until a constant weight was obtained [31].

### 2.3. Preparation of Mayonnaise

The mayonnaise ingredients included vegetable oil, egg powder, acid or enzymatic hydrolysate or hydrothermal extract, citric acid, mustard powder, salt, sugar, and water. The concentrations of additives (acid, enzymatic hydrolysates, hydrothermal extract) were determined on the basis of organoleptic indicators, primarily taste. The mayonnaise recipe is presented in Table 1.

Loose components (salt, sugar, citric acid, dry egg powder) were sieved and weighed out in accordance with recipes, then poured into food containers. Citric acid, sugar, and sodium chloride were dissolved in the required amount of water. In the case of using acid hydrolysates in the recipe, the procedure for the preparation of citric acid solution was excluded from the scheme of the technological process. The egg powder was mixed with water at a temperature of 40–45 °C and a ratio of 1:2, was heated to 60–65 °C, was held for 20–25 min for pasteurization, and then cooled to 30–40 °C. The acid hydrolysates or enzymatic hydrolysates or hydrothermal extracts, at a temperature of 55–60 °C, were mixed with the egg powder, mustard powder, and the sugar grains. The compound was stirred thoroughly and heated to 55–60 °C for 25–30 min in order to prepare the mayonnaise paste. After obtaining the mayonnaise paste and cooling it, an emulsion of mayonnaise was prepared. Vegetable oil at a temperature of 20 °C was injected into the mayonnaise paste while stirring. An acid-salt or brine solution was added after adding the whole oil dose, and the mixture was stirred for 15–20 min in order to further homogenize the emulsion. Final homogenization of the resulting mixture was carried out using a homogenizer (Kinematica, Polytron PT 45–80 GT, Switzerland). The pH value of mayonnaise was: 3.9 with acid hydrolysates, 4.1 with enzymatic hydrolysates, 4.3 with hydrothermal extracts, 3.9 with BHT, and 4.2 without additives. The samples were kept at room temperature (22 ± 2 °C). At designated times (30, 50, 70 and 90 days), the samples were taken for analyses. Mayonnaise without acid hydrolysates or enzymatic hydrolysates or hydrothermal extracts was used as a control. As a positive control, mayonnaise with the addition of BHT (Aldrich, 99.9%, analytical grade) was used.

### 2.4. Fractionation of Melanoidins

For fractionation of melanoidins, we used the method of gel-chromatography on columns with TSK-gels Toyopearl HW-40 and HW-50 (“Toyo Soda”, Japan), pre-calibrated for proteins with well-known molecular weights. The gels that were used had the following ranges for separation of molecular masses for proteins: The separation of proteins with molecular masses from 100 to 10,000 Da takes place on HW-40; and on HW-50 for proteins from 500 to 80,000 Da. A 0.2 M solution of sodium chloride and distilled water was used as eluent. Samples hydrolysates and hydrothermal extracts were applied on the column (1.2 × 35 cm, free volume 12 mL) in 0.4 mL pre-skim chloroform.

The optical density of fractions of volume 1 mL was measured using a scanning spectrophotometer UV-1800 (Shimadzu, Japan) at 400 and 420 nm wavelengths. Calculation of the melanoidins content was carried out according to formulas derived on the basis of the standard curve [32].

### 2.5. DPPH Radical Scavenging Assay

Antiradical properties were evaluated in terms of ability to interact with the stable free radical 2,2-diphenyl-1-picrylhydrazyl (DPPH) in vitro. Determination was carried out in a reaction mixture containing 3 mL of 0.3 mM DPPH in ethanol, 1 mL 50 mM tris -HCl- buffer, pH 7.4, and 1 mL of extract or hydrolysate [33]. After 30 min incubation at a temperature of 20 °C, values of optical density were recorded at λ = 517 nm. The experiments were performed on a scanning spectrophotometer UV 1800 (Shimadzu, Japan) in cuvettes (l = 1 cm) at T = 298 °K

The activity was characterized by the following indicators:-radical binding activity (RBA) was calculated by the formula
RBA (%) = (D_517_I − D_517_II)/D_517_I × 100,(1)
where D_517_I is a control, and D_517_II is a sample;-the effective concentration of substance at which 50% of free radicals *DPPH* (E_C50_) was restored;-time of recovery of half of the quantity of radical (T_EC50_), min;-antiradical efficiency (AE)—this characteristic connects the time of recovery of half of the quantity of radical (T_EC50_) to the concentration of substrate (E_C50_) necessary for this, which is calculated by the formula:
AE = 1/(E_C50_ × T_EC50_)(2)

Antiradical properties were compared with the effect of the well-known synthetic antioxidant BHT (2,6-ditretbutyl-4-methyl-phenol), which was previously purified by recrystallization from ethanol, and then the isolated crystals were dried and sublimed in a vacuum.

### 2.6. Acid and Peroxide Value (AV, PV)

The acid and peroxide values (AV, PV) in the oil samples, in the oil samples extracted from mayonnaise with a mixture of solvents (ethyl ether and chloroform) were determined according to the methods stipulated in ISO [34,35]. The peroxide values were measured spectrophotometrically at 500 nm by a UV–1800 instrument (Shimadzu, Japan). Results were expressed in mmol of oxygen per kilogram of oil. Acid value (AV) is an important indicator of vegetable oil quality. AV is expressed as the amount of KOH (in milligrams) necessary to neutralize free fatty acids contained in 1 g of oil.

### 2.7. Statistical Analysis

Results were expressed as mean values (standard deviation) (n = 9). In total, three acid hydrolysates, three enzymatic hydrolysates, and three hydrothermal extracts were obtained. Data were subjected to analysis of variance (ANOVA), and mean comparisons were carried out using the Duncan’s multiple range test (hydrolysates and hydrothermal extracts) and t-test (samples of mayonnaise with or without additives). Statistical analysis was performed using the statistical Package for Social Sciences (SPSS for windows: SPSS Inc., Chicago, IL, USA). Results with *p* < 0.05 were considered to be statistically significant.

## 3. Results and Discussion

The acid and enzymatic hydrolysates and hydrothermal extracts of motor muscle and mantle of *A. broughtonii* represent a dark brown liquid, with a specific faint odor. The content of dry substances was 15.9%–16.3% in acid hydrolysates, 8.3%–8.9% in enzymatic hydrolysates, and 5.6%–5.9% in hydrothermal extracts. The appearance of the received hydrolysates and extracts is shown in Figure 1.

Evaluation of the antiradical activity of the obtained acid and enzymatic hydrolysates and hydrothermal extracts of motor muscle and mantle of *A. broughtonii* and the well-known antioxidant BHT showed that all the studied objects have a high ability to bind radical DPPH (Table 2).

The radical-binding activity of acid and enzymatic hydrolysates and hydrothermal extracts of motor muscle and mantle of *A. broughtonii* varied within wide limits: From 55% to 89%. The results clearly indicate that the acid hydrolysate of mantle exhibited the highest radical binding activity (89%); this value is only 5% lower than for BHT. The minimum antiradical activity was shown by hydrothermal extracts of muscle, and this is characteristic for the extracts obtained from different parts of the bivalve mollusk *A. broughtonii*. Enzymatic hydrolysates possess lower antiradical activity than acid hydrolysates, but slightly higher than hydrothermal extracts. Our study showed that the method of modifying tissue of the bivalve mollusk *A. broughtonii* more significantly influences the level of manifestation of antiradical properties than the biological raw materials used. The data obtained are presented in Table 2.

The data in Table 2 demonstrate that acid hydrolysates have high antiradical efficiency (AE), and, furthermore, that acid hydrolysates from the mantle have higher values than those from muscle. The antiradical efficiency of acid hydrolysates is 35%–41% of the AE of BHT, which is a good indicator of antiradical properties. The acid hydrolysates of the mantle have 89% more AE than enzymatic hydrolysates, and 179% more than hydrothermal extracts. The AE of acid hydrolysates of muscle is 29% greater than that of enzymatic hydrolysates of muscle and 2.05 times greater than that of hydrothermal extracts (of muscle).

Similarly to our result, the protein hydrolysate obtained from the oyster *Saccostrea cucullata* exhibited 85.7% [10], the hydrolysate of the common smooth hound (*Mustelus mustelus*) exhibited 76.7% of scavenging activity [36], and protein hydrolysate of *Conus betulinus* exhibited 84% of scavenging activity [37]. By comparing the percentage of radical scavenging activity, the acid and enzymatic hydrolysates have proven to be a potential source for antiradical activity.

It is known that melanoidins and free amino acids exhibit antiradical properties. Previous research [38] has shown that acid hydrolysates of the tissues of mussels contain free amino acids, amines, dipeptides, free fatty acids, mineral substances, and high molecular substances, which are represented by two fractions: I—a molecular weight of 1500–6000 Da, and II—a weight less than 1500 Da. Moreover, the biological activity of fraction II was expressed significantly more strongly than fraction I. It was shown that fraction II comprises melanoidins having biological activity.

The fractional composition of high-molecular substances—acid and enzymatic hydrolysates and hydrothermal extracts—was researched in the present study. The quantitative proportions of fractions of melanoidins are presented in Figure 2.

High molecular substances are divided into three fractions—two of them are substances with a molecular mass of about 10,000 Da, and the third fraction is of molecular weight of about 1000 Da. The maximum content of melanoidins was found in fraction III; fractions I and II contained only a small amount of melanoidins. The acid extract from the mantle was characterized by the highest content of melanoidins in fraction III.

Synthetic antioxidants such as butylated hydroxyanisole (BHA), butylated hydroxytoluene (BHT), tert-butylhydroquinone (TBHQ), and propyl gallate are often added to foods to slow lipid oxidation.

However, concerns have been raised about their safety as additives in food. Therefore, natural antioxidants with little or no side effects are of some interest as health-friendly supplements that can preserve food quality, primarily to protect fats from oxidation [39,40,41].

It is known that protein hydrolysates and peptides may exhibit antioxidant activity against lipid and fat peroxidation [39]. It has also been shown that hydrolysates and peptides of proteins of marine origin exhibit antioxidant activity in vitro and even have higher antioxidant activity than α-tocopherol [42,43,44].

Therefore, hydrolysates and hydrothermal extracts of bivalves can be used as a functional ingredient in the food industry, in particular in the production of mayonnaise.

Synthetic antioxidants such as butylated hydroxytoluene (BHT), butylated hydroxyanisole (BHA), and ethylenediaminetetraacetic acid (EDTA) (commercial antioxidants) are widely used in mayonnaise to prevent spoilage and the appearance of a bitter taste. However, these additives have a carcinogenic effect when used in high concentrations [45]. Consumers are demonstrating a growing demand for mayonnaise with natural ingredients. Adding natural antioxidants to food oil-fat emulsion systems, such as mayonnaise, has great potential to increase their oxidative stability and will be attractive to a wider group of consumers. Also, these compounds can have an impact on health, which will position mayonnaise as a healthy and natural product [46]. Gallic acid [47], ascorbic acid [48], tocopherol [49], rosemary [50], lactoferrin [51,52], phitic acid [52], mustard [53,54] lycopene [55], ginger powder [56], grape seed extract [57], essential oils extracted from Zenyan [58], chitosan [59], and seaweed [46] have been evaluated as natural antioxidants in mayonnaise.

However, protein hydrolysates are not used as antioxidants in the composition of mayonnaise. In the present study, acidic, enzymatic hydrolysates, and hydrothermal extracts in the composition of mayonnaise significantly inhibited lipid oxidation processes, as evidenced by a decrease in peroxide value compared to the control for all studied samples, but the value of the positive control was lower (Figure 3).

The minimum peroxide value (PV) during the storage period is determined for the fat released from the mayonnaise with acidic hydrolysates, which correlates well with the high antiradical properties of the hydrolysate established earlier. The enzymatic hydrolysates and hydrothermal extracts in mayonnaise also reduce the PV of mayonnaise lipids in proportion to their antiradical properties described above. Acid and enzymatic hydrolysates and hydrothermal extracts in mayonnaise reduce PV (90 days of storage) by 27%–50%. According to the data [54], lycopene crystals containing 50 mg/kg in mayonnaise reduced the peroxide number of mayonnaise by 54% during the three-month storage period. However, a ginger powder mass fraction in mayonnaise of 1% reduced the peroxide number by only 25% during 4 months of storage [55]. Thus, acid hydrolysates reduced the peroxide number as lycopene, hydrothermal extracts as ginger powder. Mayonnaise without additives had a shelf life of no more than 90 days; the peroxide number at the end of the shelf life exceeded 20 mmol O_2_/kg. The shelf life of such mayonnaise was 17 days less than mayonnaise with additives of hydrolysates and hydrothermal extracts from the bivalve mollusk *A. broughtonii*.

The dynamics of changes of the acid value of fat isolated from mayonnaise with additives (acid, enzymatic hydrolysates and hydrothermal extracts and BHT) and without additives, during storage is presented in Figure 4.

The obtained experimental data show that during storage—in addition to the oxidation of the fat phase of mayonnaise—hydrolysis of triglycerides to form free fatty acids is observed: The AV of oil, extracted from the mayonnaise, increases during storage. Acid and enzymatic hydrolysates and hydrothermal extracts in the composition of mayonnaise slow down triglyceride hydrolysis, as evidenced by the decrease of the AV of mayonnaise compared to the control for all investigated samples. The minimum AV throughout the storage period is determined for the fat extracted from the mayonnaise with acid hydrolysate. Acid and enzymatic hydrolysates and hydrothermal extracts in mayonnaise reduce the AV (at 90 days storage) by 20%–44%.

## 4. Conclusions

The hydrolysates and hydrothermal extracts of the bivalve mollusk *A. broughtonii* possess antiradical activity, the level of manifestation of which depends on the method of biotechnological and technological processing. Assessment of antiradical activity using the DPPH radical showed that acid hydrolysates have the maximum antiradical properties, probably due to the higher content of low molecular fractions of melanoidins.

Acid and enzymatic hydrolysates, and hydrothermal extracts in the mayonnaise composition slow down the processes of lipid oxidation and hydrolysis of triglycerides. Acid hydrolysates play the most significant role in reducing the speed of oxidation and hydrolysis. The results of this study can be practically applied in the production of mayonnaise and fat-containing sauces. However, the hydrolysates and extracts were not able to prevent the formation of peroxides as is the case for EDTA, which is a much better antioxidant in mayonnaise than BHT. Moreover, peroxides do not have any effect on off-flavors. To make a more complete evaluation of the effect of the hydrolysates and extracts determination of secondary lipid oxidation products such as aldehydes and ketones is necessary and preferably sensory analysis should also be done.

## Figures and Tables

**Figure 1 foods-09-00304-f001:**
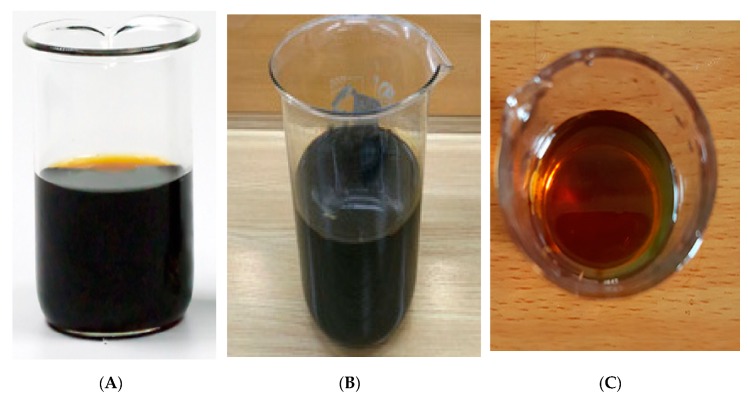
Appearance of hydrolysates and extracts (**A**—acid hydrolysate (AH) muscle, **B**— enzymatic hydrolysates (EH) mantle, **C**—hydrothermal extracts (HTE) muscle).

**Figure 2 foods-09-00304-f002:**
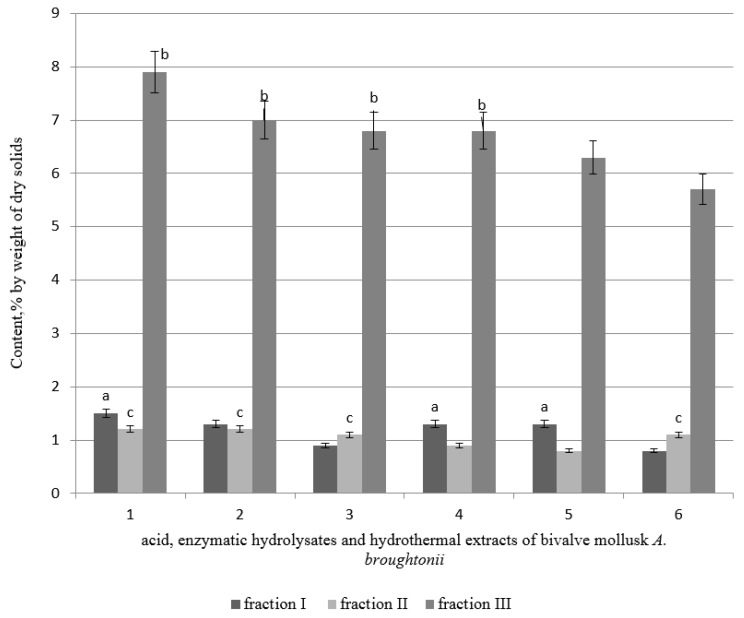
The proportion of melanoidins fractions in hydrolysates and extracts of bivalve mollusk *A. broughtonii* (1—AH muscle, 2—AH mantle, 3—enzymatic hydrolysate (EH) muscle, 4—EH mantle, 5—HTE muscle, 6—HTE mantle). Data are mean ± standard deviation (n = 9). letters indicate which fraction has the highest content of dry solids. a—fraction I versus fraction II (*p* < 0.05). b—fraction I versus fraction III (*p* < 0.05). c—fraction II versus fraction III (*p* < 0.05).

**Figure 3 foods-09-00304-f003:**
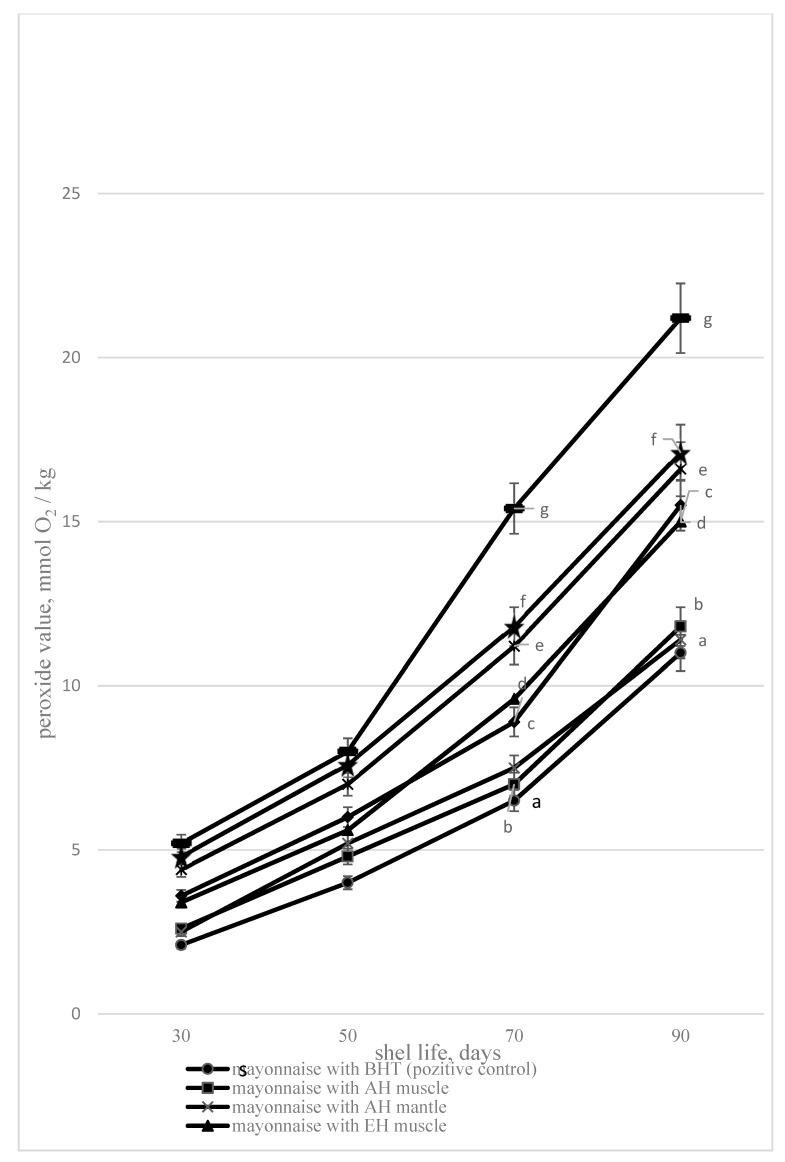
The peroxide value depending on the storage duration 1—mayonnaise with AH muscle, 2—mayonnaise with EH muscle, 3—mayonnaise with HTE muscle, 4—mayonnaise without additives (negative control), 5—mayonnaise with butylated hydroxytoluene (BHT) (positive control), 6—mayonnaise with AH mantle, 7—mayonnaise with EH mantle, 8—mayonnaise with HTE mantle). Data are mean ± standard deviation (n = 9). a—mayonnaise with AH muscle versus mayonnaise without additives (negative control); b—mayonnaise with AH mantle versus mayonnaise without additives (negative control); c—mayonnaise with EH mantle versus mayonnaise without additives (negative control); d—mayonnaise with EH muscle versus mayonnaise without additives (negative control), e—mayonnaise with HTE muscle versus mayonnaise without additives (negative control), f—mayonnaise with HTE mantle versus mayonnaise without additives (negative control), and g—mayonnaise with BHT (positive control) versus mayonnaise without additives (negative control) (*p* < 0.05).

**Figure 4 foods-09-00304-f004:**
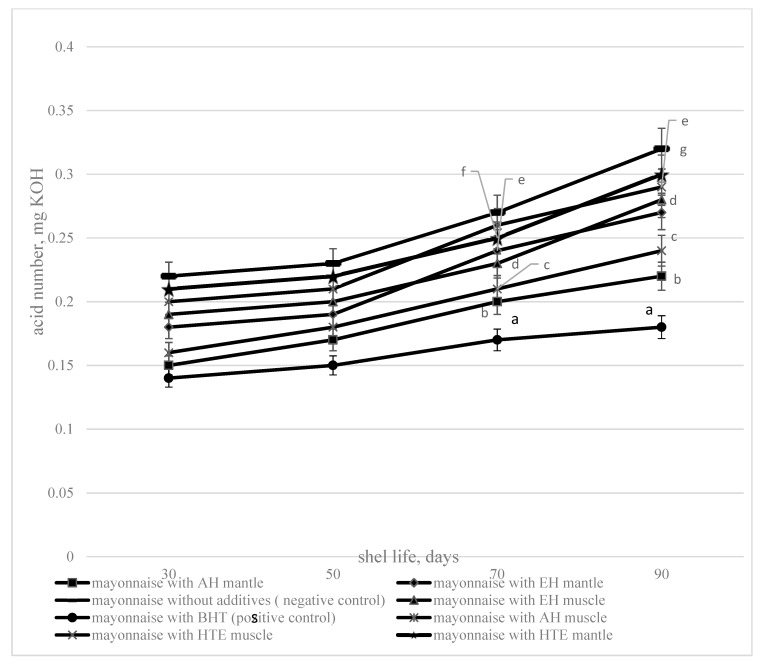
The acid value depending on the storage duration (1—mayonnaise with AH muscle, 2—mayonnaise with EH muscle, 3—mayonnaise with HTE muscle, 4—mayonnaise without additives (negative control), 5—mayonnaise with BHT (positive control), 6—mayonnaise with AH mantle, 7—mayonnaise with EH mantle, 8—mayonnaise with HTE mantle). Data are mean ± standard deviation (n = 9). a—mayonnaise with AH muscle versus mayonnaise without additives (negative control); b—mayonnaise with AH mantle versus mayonnaise without additives (negative control); c—mayonnaise with EH mantle versus mayonnaise without additives (negative control); d—mayonnaise with EH muscle versus mayonnaise without additives (negative control), e—mayonnaise with HTE muscle versus mayonnaise without additives (negative control), f—mayonnaise with HTE mantle versus mayonnaise without additives (negative control), and g—mayonnaise with BHT (positive control) versus mayonnaise without additives (negative control) (*p* < 0.05)

**Table 1 foods-09-00304-t001:** Composition of mayonnaise with hydrolysates and hydrothermal extracts from mollusks (*A. broughtonii*).

Component	Content, g/100 g		
	Mayonnaise with Acid Hydrolysate	Mayonnaise with Enzymatic Hydrolysate	Mayonnaise with Hydrothermal Hydrolysate
Vegetable oil	67	67	67
Egg powder	5	5	5
Acid hydrolysate	12	-	-
Enzymatic hydrolysate	-	18	-
Hydrothermal extract	-	-	20
Citric acid	-	0.4	0.4
Mustard powder	0.75	0.75	0.75
Salt	1.0	1.0	1.0
Sugar	1.5	1.5	1.5
Water	12.75	18.35	20.35
Total	100	100	100

**Table 2 foods-09-00304-t002:** Antiradical activity of hydrolysates and extracts from mollusk *A. broughtonii*.

An Object	RBA%			EC_50_, mkg/mL				T_EC50_, min				AE, mkg/l∙s × 10^−2^			
	Subset for a = 0.05			Subset for a = 0.05				Subset for a = 0.05				Subset for a = 0.05			
	1	2	3	1	2	3	4	1	2	3	4	1	2	3	4
AH muscle	86.2 ± 3.21			16.1 ± 0.80				13.9 ± 0.65				0.45 ± 0.02			
AH mantle	89.3 + 4.02			15.5 ± 0.75				12.1 ± 0.60				0.53 ± 0.01			
EH muscle		70.5 ± 2.89				19.7 ± 0.94			14.5 ± 0.72				0.35 ± 0.01		
EH mantle		68.4 ± 2.95				22.6 ± 1.10			16.0 ± 0.80				0.28 ± 0.01		
HTE muscle			55.2 ± 2.17		27.5 ± 1.35					16.4 ± 0.80				0.22 ± 0.01	
HTE mantle			59.3 ± 2.58		30.2 ± 1.50					17.3 ± 0.81				0.19 ± 0.01	
BHT	94.3 ± 4.08						8.75 ± 0.41				7.00 ± 0.32				1.6 ± 0.07
Sig.	1.00	1.00	1.00	1.00	1.00	1.00		1.00	1.00	1.00		1.00	1.00	1.00	1.00

Means for groups in homogenous groups that are not statistically different are displayed. Data are mean ± standard deviation (n = 9).

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
