# Peer review of "Antiradical Activity of Hydrolysates and Extracts from Mollusk A. broughtonii and Practical Application to the Stabilization of Lipids"

_foods, 2020, doi:10.3390/foods9030304_

Round 1
Reviewer 1 Report
The manuscript entitled “Antiradical activity of hydrolysates and extracts from mollusc A. broughtonii and practical application to the stabilisation of lipids” describes the in vitro antioxidant activity of hydrolysates and extracts obtained from a bivalve mollusc found in the Sea of Japan, and its application as an additive on a food product to increase the preservation capacity. The aim and scope fit perfectly within the special issue "Advance in Recovery and Application of Bioactive Compounds from Seafood". However, FOODS is a reference journal in the field, and, thus, high-quality scientific evidence, as well as suitable writing, must be provided when submitting articles.
Overall, the manuscript needs to review the wording. Please improve English of the manuscript as it is sometimes difficult to follow the descriptions throughout the text. There seems to be a general lack of order.
Introduction:
In order to clarify the necessity to assess the melanoidin content and fractionation, it would be recommended to include in the Introduction section some explanation about the relationship with the antioxidant activity of the molluscs/seafood. Also, any specific information about the mollusc species A. broughtonii is encouraged.
Line 37: Please, add a reference.
Line 47: antioxidants
Line 45 – 50: statements done by the authors are quite general. Please, improve the text with concise data and include some references, especially for “Natural antioxidants are more effective at stabilising the quality of food fat.”
Material and methods
Line 56 – 60: it is suggested to reject the “Enzymes and other chemicals” subsection. All information about the reagents and the equipment employed should be mentioned throughout the Materials and methods section, where they are mentioned.
Line 61-63: Please, describe better the sampling procedure, including more information about the region of capture, number of samples, transport, pre-treatments, and procedures applied.
Line 80: The method included of elaboration of mayonnaise (ТР10.04.40.65-88) links to a website in Russian. Please, avoid this reference and describe further and better the processing, or include any reference available for an English-speaking audience.
Line 79: Preparation of mayonnaise. In the Results and discussion sections is included data obtained from a shelf-life study on the mayonnaise for up to 90 days. However, no information about the shelf-life conditions are described in the M&M section. Please, clarify this issue.
Line 96: if possible, it would be encouraged the term BHT instead of ionol, as it is more commonly used in the food sector. Also, provide more information about the BHT (concentration, purity).
Line 97 (Table 1): please, avoid the use of commas for decimal figures. Also, if possible, avoid the use of % for quantifying; it is preferable to employ units such as g/100g or ml/100mL. Include the composition of negative and positive mayonnaise. What were the criteria used to determine the % of the addition of each hydrolysate/extract? According to the 2.3 section, different concentrations of the sample were used to do the extractions, but no criteria were specified to assess the final concentrations. Please, explain better.
Line 111. DPPH radical scavenging assay. The reference to the method employed is common and well-accepted to assess the lipophilic antioxidant activity in the food sector. However, it is poorly described and may create confusion with an audience not used to it. Please, improve it. If it is possible, it would be highly encouraged to include extra antioxidant analyses focused on the hydrophilic (i.e. ABTS) and reducing (i.e. FRAP) antioxidant activity to obtain a complete view of the antioxidant activity of the extracts.
Line 129: Peroxide value. It is said that the PV was measured in fat samples isolated from mayonnaise. Please, indicate the method employed for extracting the fats from the mayonnaise.
Line 136: gm not suitable for grams.
Line 134-139: the method is poorly described. Please, improve the wording.
Line 140 Statistical analysis. Further description of the statistical analysis employed is required. The authors need to describe how were the replicates obtained (how many bivalves employed for each extraction, how many extractions were performed, how many samples of mayonnaise were employed at each control point, etc.).
Besides, I would suggest replacing the statistical analysis employed. In order to elucidate the effect of the extraction method (acid, enzymatic, hydrothermal) and the section of the mollusc (mantle, muscle) and potential interactions, a GLM procedure should be carried out. The p values of the analysis should also be included in the tables, that need to be improved, also in the presentation. Figure 1 is irrelevant, and the data presented should be included with the other antioxidant information in the corresponding amended table.
Results and discussion
I strongly suggest rewriting the Results and Discussion sections separately to a more precise explanation of the findings. The scientific background of the Discussion would need to be improved.
Line 147-148: Apart from the subjective description of the hydrolysates/extracts, it would be suitable to include images of the samples obtained.
Line 148-149: “The content of dry substances was 15.9-16.3% in acid hydrolysates, 8.3-8.9% in enzymatic hydrolysates and 5.6-5.9% in hydrothermal extracts.” No reference to the method employed to quantify the content of dry substances is included in the M&M section.
Line 166: check the use of italics.
Line 196: Figure 2 needs to be improved. The statistical contrasts described in the footnote are not described in the M&M section.
Line 213: check the use of italics.
Line 234: include error bars in the graph. Check the letter superscripts.
Line 241: “(…) which correlates well with the high antiradical properties of the hydrolysate established earlier”. No correlation analysis was performed to include this statement.
Author Response
First of all, we would like to thank the reviewers for valuable comments that allowed us to increase the value of our manuscript.
Below are the answers to the review
Open Review
(x) I would not like to sign my review report
( ) I would like to sign my review report
English language and style
( ) Extensive editing of English language and style required
(x) Moderate English changes required
( ) English language and style are fine/minor spell check required
( ) I don't feel qualified to judge about the English language and style
|
Yes |
Can be improved |
Must be improved |
Not applicable |
|
|
Does the introduction provide sufficient background and include all relevant references? |
( ) |
( ) |
(x) |
( ) |
|
Is the research design appropriate? |
( ) |
( ) |
(x) |
( ) |
|
Are the methods adequately described? |
( ) |
(x) |
( ) |
( ) |
|
Are the results clearly presented? |
( ) |
( ) |
(x) |
( ) |
|
Are the conclusions supported by the results? |
( ) |
(x) |
( ) |
( ) |
Comments and Suggestions for Authors
The manuscript entitled “Antiradical activity of hydrolysates and extracts from mollusc A. broughtonii and practical application to the stabilisation of lipids” describes the in vitro antioxidant activity of hydrolysates and extracts obtained from a bivalve mollusc found in the Sea of Japan, and its application as an additive on a food product to increase the preservation capacity. The aim and scope fit perfectly within the special issue "Advance in Recovery and Application of Bioactive Compounds from Seafood". However, FOODS is a reference journal in the field, and, thus, high-quality scientific evidence, as well as suitable writing, must be provided when submitting articles.
Overall, the manuscript needs to review the wording. Please improve English of the manuscript as it is sometimes difficult to follow the descriptions throughout the text. There seems to be a general lack of order.
Introduction:
In order to clarify the necessity to assess the melanoidin content and fractionation, it would be recommended to include in the Introduction section some explanation about the relationship with the antioxidant activity of the molluscs/seafood. Also, any specific information about the mollusc species A. broughtonii is encouraged.
Line 37: Please, add a reference. - corrected
Line 47: antioxidants - corrected
Line 45 – 50: statements done by the authors are quite general. Please, improve the text with concise data and include some references, especially for “Natural antioxidants are more effective at stabilising the quality of food fat.” - corrected
Material and methods
Line 56 – 60: it is suggested to reject the “Enzymes and other chemicals” subsection. All information about the reagents and the equipment employed should be mentioned throughout the Materials and methods section, where they are mentioned. - corrected
Line 61-63: Please, describe better the sampling procedure, including more information about the region of capture, number of samples, transport, pre-treatments, and procedures applied. –
corrected
Line 80: The method included of elaboration of mayonnaise (ТР10.04.40.65-88) links to a website in Russian. Please, avoid this reference and describe further and better the processing, or include any reference available for an English-speaking audience. - corrected
Line 79: Preparation of mayonnaise. In the Results and discussion sections is included data obtained from a shelf-life study on the mayonnaise for up to 90 days. However, no information about the shelf-life conditions are described in the M&M section. Please, clarify this issue. corrected
Line 96: if possible, it would be encouraged the term BHT instead of ionol, as it is more commonly used in the food sector. Also, provide more information about the BHT (concentration, purity). - corrected
Line 97 (Table 1): please, avoid the use of commas for decimal figures. Also, if possible, avoid the use of % for quantifying; it is preferable to employ units such as g/100g or ml/100mL. Include the composition of negative and positive mayonnaise. What were the criteria used to determine the % of the addition of each hydrolysate/extract? According to the 2.3 section, different concentrations of the sample were used to do the extractions, but no criteria were specified to assess the final concentrations. Please, explain better. corrected
Line 111. DPPH radical scavenging assay. The reference to the method employed is common and well-accepted to assess the lipophilic antioxidant activity in the food sector. However, it is poorly described and may create confusion with an audience not used to it. Please, improve it. If it is possible, it would be highly encouraged to include extra antioxidant analyses focused on the hydrophilic (i.e. ABTS) and reducing (i.e. FRAP) antioxidant activity to obtain a complete view of the antioxidant activity of the extracts.
We think the method is described correctly and sufficiently..
Line 129: Peroxide value. It is said that the PV was measured in fat samples isolated from mayonnaise. Please, indicate the method employed for extracting the fats from the mayonnaise. corrected
Line 136: gm not suitable for grams. corrected
Line 134-139: the method is poorly described. Please, improve the wording. corrected
Line 140 Statistical analysis. Further description of the statistical analysis employed is required. The authors need to describe how were the replicates obtained (how many bivalves employed for each extraction, how many extractions were performed, how many samples of mayonnaise were employed at each control point, etc.).
Besides, I would suggest replacing the statistical analysis employed. In order to elucidate the effect of the extraction method (acid, enzymatic, hydrothermal) and the section of the mollusc (mantle, muscle) and potential interactions, a GLM procedure should be carried out. The p values of the analysis should also be included in the tables, that need to be improved, also in the presentation. Figure 1 is irrelevant, and the data presented should be included with the other antioxidant information in the corresponding amended table.
The statistical methods used were consulted with experts in this field and in their opinion our approach is acceptable.
Results and discussion
I strongly suggest rewriting the Results and Discussion sections separately to a more precise explanation of the findings. The scientific background of the Discussion would need to be improved.
Line 147-148: Apart from the subjective description of the hydrolysates/extracts, it would be suitable to include images of the samples obtained. corrected
Line 148-149: “The content of dry substances was 15.9-16.3% in acid hydrolysates, 8.3-8.9% in enzymatic hydrolysates and 5.6-5.9% in hydrothermal extracts.” No reference to the method employed to quantify the content of dry substances is included in the M&M section. corrected
Line 166: check the use of italics. corrected
Line 196: Figure 2 needs to be improved. The statistical contrasts described in the footnote are not described in the M&M section. corrected
Line 213: check the use of italics. corrected
Line 234: include error bars in the graph. Check the letter superscripts.
Line 241: “(…) which correlates well with the high antiradical properties of the hydrolysate established earlier”. No correlation analysis was performed to include this statement. corrected
Submission Date
29 December 2019
Date of this review
07 Jan 2020 13:38:34
Reviewer 2 Report
The paper has a innovactive character, but it needs of major revisions concerning the presentation of paper. The authors should better mark in the Conclusion the potental and practical applications of this research. The introduction should be improved by adding a detailed description of main bioactive components of mollusk and mark the updated state of research on antioxidant properties and add related refrences. The authors should add more details and indicate the number of samples and add a scheme for sampling in subparagraph Material of the bivalve mollusk. Detail on sampling and procedures should be added also for subparagraph Preparation of mayonnaise.
The format of Figure 1 and 2 should be checked.
Check the Table 1 decimal places.
Table 2 should be redesigned and better discussed in the text.
Lines 166-168 should better clarified. In lines 181-185 the authors should better describe the matter in the citter references.
Author Response
Comments and Suggestions for Authors
The paper has a innovactive character, but it needs of major revisions concerning the presentation of paper.
The authors should better mark in the Conclusion the potental and practical applications of this research. - corrected
The introduction should be improved by adding a detailed description of main bioactive components of mollusk and mark the updated state of research on antioxidant properties and add related refrences. - corrected
The authors should add more details and indicate the number of samples and add a scheme for sampling in subparagraph Material of the bivalve mollusk. Detail on sampling and procedures should be added also for subparagraph Preparation of mayonnaise. corrected
The format of Figure 1 and 2 should be checked. - what does it mean?
Check the Table 1 decimal places. corrected
Table 2 should be redesigned and better discussed in the text. corrected
Lines 166-168 should better clarified. In lines 181-185 the authors should better describe the matter in the citter references. corrected
Round 2
Reviewer 1 Report
I would like to thank the authors for the changes applied according to the corrections suggested. However, most of them were minor changes that do not change the main problems found in the manuscript.
First of all, please, review the English language.
Secondly, the experimental design is still poorly described, and I do not agree with the statistical analysis approach:
"A mollusc was collected once every month..." for a total of 5 months. Then, 5 molluscs were collected in total? All analyses were done using pooled samples - how were the replicates selected? I tend to think that the replicates included in the study (n=3) were only laboratory replicates and not experimental replicates. How many hydrolysis/extractions were performed for each sample? Regarding the mayonnaise, again, how many samples were analysed per sampling time? I assume that a pasteurisation step was applied to the mayonnaise at the end of the process. Describe it. Also, the recipient and light conditions in which they were kept during the study. The statistical analysis performed does not help to compare the differences between the different extraction methods (acid, enzymatic, thermal) and sections of the mollusc (muscle, mantle). I highly encourage to change the approach. The Figures and Tables confirmed the lack of a suitable statistical approach and must be improved. The superscripts do not give information about the differences between the treatments, and they are not adequately employed (Fig. 3 and 4: the order of the letters does not correspond with the differences between samples. The letter "d", for instance, represents both the higher and lower treatment). Besides, no error bars are included in the graphs showing the SD of the samples at each sampling point. Also, the use of the scientific notation in table 2 (AE column) is entirely unnecessary.Other major issues:
There is a lack of scientific background in the discussion. I strongly suggest rewriting the Results and Discussion sections separately to a more precise explanation of the findings. During the description of the PV and AV methods, the new references included (ISO methods) do not correspond with the references employed in the previous version. Please, clarify this issue.
Author Response
First of all, please, review the English language.
The text of the manuscript was checked twice by an English native speaker with university education. American spelling has also been corrected. The person performing the check is now an English proofreader in the journal Problems of Forensic Sciences.
If the reviewer has comments on the English language, our proofreader asks you to send examples of language incorrectness
Secondly, the experimental design is still poorly described, and I do not agree with the statistical analysis approach:
A mollusc was collected once every month..." for a total of 5 months. Then, 5 molluscs were collected in total? All analyses were done using pooled samples - how were the replicates selected? I tend to think that the replicates included in the study (n=3) were only laboratory replicates and not experimental replicates. How many hydrolysis/extractions were performed for each sample? Regarding the mayonnaise, again, how many samples were analysed per sampling time? I assume that a pasteurisation step was applied to the mayonnaise at the end of the process. Describe it. Also, the recipient and light conditions in which they were kept during the study.
All the doubts of the reviewer were clarified in the manuscript text.
The statistical analysis performed does not help to compare the differences between the different extraction methods (acid, enzymatic, thermal) and sections of the mollusc (muscle, mantle). I highly encourage to change the approach. The Figures and Tables confirmed the lack of a suitable statistical approach and must be improved. The superscripts do not give information about the differences between the treatments, and they are not adequately employed (Fig. 3 and 4: the order of the letters does not correspond with the differences between samples. The letter "d", for instance, represents both the higher and lower treatment). Besides, no error bars are included in the graphs showing the SD of the samples at each sampling point. Also, the use of the scientific notation in table 2 (AE column) is entirely unnecessary.
Additional statistical analysis was carried out and the result was included in the manuscript.
Other major issues:
There is a lack of scientific background in the discussion. I strongly suggest rewriting the Results and Discussion sections separately to a more precise explanation of the findings. During the description of the PV and AV methods, the new references included (ISO methods) do not correspond with the references employed in the previous version. Please, clarify this issue.
Additional references from the pistols have been added and the doubts raised have been clarified.
Due to the limited amount of valuable tests, the results were not separated from the tests
Reviewer 2 Report
the article has been improved as requested now it is ready to be publishedAuthor Response
First of all, please, review the English language.
The text of the manuscript was checked twice by an English native speaker with university education. American spelling has also been corrected. The person performing the check is now an English proofreader in the journal Problems of Forensic Sciences.
If the reviewer has comments on the English language, our proofreader asks you to send examples of language incorrectness
Secondly, the experimental design is still poorly described, and I do not agree with the statistical analysis approach:
A mollusc was collected once every month..." for a total of 5 months. Then, 5 molluscs were collected in total? All analyses were done using pooled samples - how were the replicates selected? I tend to think that the replicates included in the study (n=3) were only laboratory replicates and not experimental replicates. How many hydrolysis/extractions were performed for each sample? Regarding the mayonnaise, again, how many samples were analysed per sampling time? I assume that a pasteurisation step was applied to the mayonnaise at the end of the process. Describe it. Also, the recipient and light conditions in which they were kept during the study.
All the doubts of the reviewer were clarified in the manuscript text.
The statistical analysis performed does not help to compare the differences between the different extraction methods (acid, enzymatic, thermal) and sections of the mollusc (muscle, mantle). I highly encourage to change the approach. The Figures and Tables confirmed the lack of a suitable statistical approach and must be improved. The superscripts do not give information about the differences between the treatments, and they are not adequately employed (Fig. 3 and 4: the order of the letters does not correspond with the differences between samples. The letter "d", for instance, represents both the higher and lower treatment). Besides, no error bars are included in the graphs showing the SD of the samples at each sampling point. Also, the use of the scientific notation in table 2 (AE column) is entirely unnecessary.
Additional statistical analysis was carried out and the result was included in the manuscript.
Other major issues:
There is a lack of scientific background in the discussion. I strongly suggest rewriting the Results and Discussion sections separately to a more precise explanation of the findings. During the description of the PV and AV methods, the new references included (ISO methods) do not correspond with the references employed in the previous version. Please, clarify this issue.
Additional references from the pistols have been added and the doubts raised have been clarified.
Due to the limited amount of valuable tests, the results were not separated from the tests